# Systems biology analysis uncovers a ROS-associated gene signature and immunomodulatory role of CLEC4E in ischemic stroke

Lifang Yang[1], Tianyu Liang[2], Xiaodi Ding🆔[1]*

**1** Center for Rehabilitation Medicine, Rehabilitation & Sports Medicine Research Institute of Zhejiang Province, Department of Rehabilitation Medicine, Zhejiang Provincial People's Hospital(Affiliated People's Hospital), Hangzhou Medical College, Hangzhou, Zhejiang, China, **2** Emergency and Critical Care Center, Intensive Care Unit, Zhejiang Provincial People's Hospital, Affiliated People's Hospital, Hangzhou Medical College, Hangzhou, Zhejiang, China

* Qiwenlong11588@126.com

## Abstract

### Background

Reactive oxygen species (ROS) are critically implicated in ischemic stroke (IS), yet the transcriptional networks and predictive biomarkers underlying ROS dysregulation remain incompletely understood.

### Methods

We integrated two independent microarray cohorts (GSE58294 and GSE16561) to comprehensively analyze ROS-related pathways in IS. Single-sample gene set enrichment analysis (ssGSEA) was used to quantify pathway activity, and weighted gene co-expression network analysis (WGCNA) identified modules associated with ROS dysregulation. Functional enrichment and protein-protein interaction (PPI) network analyses characterized the biological functions of module genes. Elastic Net regression modeling, receiver operating characteristic (ROC) analysis, calibration, and decision curve analysis (DCA) were employed to construct and validate a predictive risk score model. SHapley Additive exPlanations (SHAP) analysis was further applied to interpret gene contributions. Immune cell infiltration was assessed using multiple algorithms, and CLEC4E, the top-ranked gene, was functionally investigated through single-gene GSEA. The OGD/R-treated SH-SY5Y cells and mouse ischemia-reperfusion (I/R) models were established for in vivo and in vitro validation.

### Results

ROS-related pathways were consistently upregulated in IS across both cohorts. WGCNA revealed a robust ROS-associated module (brown module), enriched in

**Data availability statement:** Data is provided within the manuscript or Supporting Information files.

**Funding:** This work was supported by the Zhejiang Provincial Administration of Traditional Chinese Medicine plans (Grant No. C-2025-W219). The funders had no role in study design, data collection and analysis, decision to publish, or preparation of the manuscript.

**Competing interests:** The authors have declared that no competing interests exist.

**Abbreviations:** ROS: Reactive oxygen species; IS: Ischemic stroke; ssGSEA: Single-sample gene set enrichment analysis; WGCNA: Weighted gene co-expression network analysis; PPI: Protein-protein interaction; ROC: Receiver operating characteristic; DCA: Decision curve analysis; SHAP: SHapley Additive exPlanations; I/R: Ischemia-reperfusion; GEO: Gene Expression Omnibus; NES: Normalized enrichment scores; GO: Gene Ontology; KEGG: Kyoto Encyclopedia of Genes and Genomes; BP: Biological process; CC: Cellular component; MF: Molecular function; FDR: False discovery rate; AUC: Area under the curve; OOF: Out-of-fold; C-index: Concordance index; EDA: Edaravone; HE: Hematoxylin and Eosin; MCAO/R: Middle cerebral artery occlusion and reperfusion; qRT-PCR: Quantitative reverse transcription polymerase chain reaction; cDNA: Complementary DNA; PVDF: Polyvinylidene difluoride; ECL: Enhanced chemiluminescence.

immune activation and inflammatory signaling processes. Elastic Net regression identified seven key genes (CLEC4E, SLC8A1, HIST1H4H, BMX, MCEMP1, KREMEN1, ZFP36L2) with strong predictive ability (AUC = 0.81–0.86 across datasets). SHAP analysis highlighted CLEC4E as the most influential contributor, positively associated with IS risk. Immune deconvolution indicated that CLEC4E expression was negatively correlated with B- and T-cell infiltration, while functional analysis linked it to MAPK signaling, RNA degradation, and neutrophil activation pathways. Finally, CLEC4E was significantly elevated in IS. Knockdown of CLEC4E could alleviate the effect of OGD/R on SH-SY5Y cells.

## Conclusions

Our study demonstrates pervasive activation of ROS-related transcriptional programs in IS and identifies a novel seven-gene signature predictive of disease risk. Among these, CLEC4E emerges as a key mediator connecting ROS dysregulation to immune infiltration and inflammatory signaling, providing new insights into IS pathophysiology and potential therapeutic targets.

## Introduction

The burden of ischemic stroke (IS) continues to rise globally, accounting for approximately 70% of all stroke cases and contributing substantially to long-term disability and mortality [1,2]. The incidence of IS continues to rise due to population aging and the persistence of key risk factors, including hypertension, diabetes, smoking, and obesity [3,4]. Despite advances in reperfusion therapy and secondary prevention, effective biomarkers for early diagnosis and prognosis remain limited, underscoring the urgent need to elucidate novel molecular mechanisms driving IS pathogenesis [5].

Reactive oxygen species (ROS) play a pivotal role in cerebral ischemia-reperfusion injury, where their excessive accumulation induces oxidative stress and damages lipids, proteins, and nucleic acids, ultimately leading to neuronal apoptosis and necrosis [6,7]. ROS not only disrupt redox homeostasis but also amplify inflammatory cascades, activate microglia, and impair blood-brain barrier integrity, thereby exacerbating secondary brain injury [8,9]. Although antioxidants have been tested extensively in preclinical and clinical studies, most therapeutic interventions have failed to translate into clinical benefit, suggesting that a more comprehensive understanding of ROS-related molecular networks is required [10,11].

In recent years, bioinformatics has emerged as a powerful approach to dissect complex disease mechanisms and identify novel biomarkers at a systems level [12,13]. Through transcriptomic profiling, network-based analyses, and machine learning algorithms, researchers can integrate large-scale datasets to identify disease-associated gene modules, predict clinical outcomes, and highlight potential therapeutic targets [14,15]. These strategies are particularly valuable in stroke research, where heterogeneous molecular mechanisms complicate biomarker discovery and therapeutic development.

In this study, we systematically investigated ROS-related pathways in ischemic stroke using two independent GEO cohorts (GSE58294 and GSE16561). We employed single-sample gene set enrichment analysis (ssGSEA) and weighted gene co-expression network analysis (WGCNA) to identify ROS-associated gene modules, constructed a predictive gene signature using Elastic Net regression and SHapley Additive exPlanations (SHAP) interpretation, and explored the immunological role of CLEC4E through immune infiltration and enrichment analyses. Our integrative approach aimed to provide new insights into the molecular mechanisms of ROS in IS and highlight potential targets for future therapeutic interventions.

## Materials and methods

### 2.1 Data acquisition and preprocessing

Public transcriptomic datasets related to ischemic stroke (IS) were obtained from the Gene Expression Omnibus (GEO) database [16]. Specifically, two independent microarray cohorts were included: GSE58294 [17], comprising peripheral blood samples from 69 IS patients (collected at three time points: within 3 hours, at 5 hours, and at 24 hours post-onset) and 23 healthy controls, and GSE16561 [18], comprising whole-blood samples from 39 IS patients and 24 healthy controls. The use of these de-identified public datasets is consistent with ethical guidelines as all original studies obtained appropriate approvals. Raw data were background-corrected and normalized using the limma package in R. Probes were mapped to official gene symbols based on platform annotation files; for multiple probes corresponding to the same gene, the average expression value was retained. Quality control was performed using principal component analysis (PCA), and all samples showed consistent clustering within their respective groups.

### 2.2 Identification of ROS-related gene sets

To systematically characterize ROS-associated biological programs, we searched the Molecular Signatures Database (MSigDB) [19] and curated five ROS-related pathways, including HALLMARK REACTIVE OXYGEN SPECIES PATHWAY, GOBP CELLULAR RESPONSE TO REACTIVE OXYGEN SPECIES, GOBP REACTIVE OXYGEN SPECIES BIOSYNTHETIC PROCESS, GOBP REACTIVE OXYGEN SPECIES METABOLIC PROCESS, and GOBP RESPONSE TO REACTIVE OXYGEN SPECIES. These gene sets were selected based on the following considerations: (a) They are all derived from the authoritative MSigDB database and feature well-defined biological annotations; (b) They collectively cover distinct dimensions of ROS biology—such as biosynthesis, metabolism, and cellular response—thus providing a comprehensive reflection of ROS pathway activity; (c) Each of these gene sets has been previously implicated in ischemic diseases by multiple independent studies. These gene sets were used to represent different aspects of ROS activity in IS.

### 2.3 Pathway activity quantification and differential analysis

The ssGSEA was applied to quantify the enrichment scores of each ROS-related pathway at the individual sample level. The analysis was conducted using the "GSVA" package in R software (version 4.5.1). For each patient, normalized enrichment scores (NES) were computed, and the Z-score transformation was applied across samples to enable direct comparison. To evaluate the transcriptional alterations of ROS pathways, the ssGSEA scores of IS samples were compared with those of normal controls in both GSE58294 and GSE16561 cohorts.

### 2.4 WGCNA

By selecting the most core and directly ROS-related gene sets from MSigDB and rigorously filtering them, ROS-related genes were derived. These genes from the GSE58294 cohort were then subjected to weighted gene co-expression network analysis (WGCNA) using the "WGCNA" package in R. A soft-thresholding power of 12 was chosen to satisfy the scale-free topology criterion and ensure a biologically meaningful network. Modules were identified via hierarchical clustering using the following parameters: a minimum module size of 30 genes, and a merge cut height of 0.25. Genes were

clustered into distinct modules based on topological overlap, and module eigengenes were subsequently calculated to represent the expression profile of each module.

## 2.5  Module-trait correlation and preservation analysis

Pearson correlation analysis was used to assess the correlation between module eigengenes and ROS pathway scores; modules with the strongest positive correlation were defined as functionally relevant. To verify the robustness of co-expression modules identified in the GSE58294 cohort, we performed module preservation analysis via the modulePreservation function. The GSE58294 dataset was split into a training set (75%) and a test set (25%), with median rank and Z-summary calculated to quantify module preservation.

## 2.6  Functional enrichment analysis

Genes from the brown module were ranked according to differential expression between IS and normal samples, and the top 200 upregulated and top 200 downregulated genes were selected for further functional analysis. Protein-protein interaction (PPI) networks were constructed using the STRING database with a confidence score > 0.7. To explore biological functions, Gene Ontology (GO) and Kyoto Encyclopedia of Genes and Genomes (KEGG) pathway enrichment analyses were performed separately for upregulated and downregulated genes using the "clusterProfiler" R package, covering biological process (BP), cellular component (CC), and molecular function (MF) categories [20]. Enrichment results with adjusted $P < 0.05$ and false discovery rate (FDR) < 0.05 were considered significant [21].

## 2.7  Elastic net regression modeling

The top 10 upregulated and top 10 downregulated genes from the brown module (20 genes in total) were selected as candidate predictors. Pairwise correlations among these genes were first assessed in the GSE58294 cohort to examine their co-expression relationships. To construct a predictive model for IS, Elastic Net regression was performed using the "glmnet" package in R, with 10-fold cross-validation applied to optimize the penalty parameters (λ). Genes with nonzero coefficients in the optimal model were retained as key predictors. The discriminative ability of the model was then evaluated in the GSE58294 training set and validated in the independent GSE16561 dataset by calculating predicted probabilities for IS versus control samples.

## 2.8  Model performance evaluation

The predictive ability of each candidate gene was first evaluated by receiver operating characteristic (ROC) curve analysis, and the area under the curve (AUC) was calculated in both the GSE58294 and GSE16561 cohorts using the "pROC" R package. To assess the overall performance of the composite risk score model and avoid overfitting, repeated cross-validation was performed, and the distribution of out-of-fold (OOF) AUC values was estimated. Model calibration was examined by comparing predicted probabilities with observed outcomes, and the concordance index (C-index) was computed. Finally, decision curve analysis (DCA) was conducted using the "rmda" R package to evaluate the net clinical benefit of the risk score compared with "treat all" or "treat none" strategies across a range of threshold probabilities.

## 2.9  SHapley additive exPlanations analysis

To improve the interpretability of the predictive model, SHAP analysis was performed using the "kernelshap" R package. SHAP values were calculated to quantify the contribution of each gene to the risk score prediction in individual samples. Feature importance was assessed by the mean absolute SHAP value across all samples, and summary plots were generated to visualize the overall impact and direction of gene expression on model output. Dependence plots were used to

examine the relationship between gene expression levels and corresponding SHAP values. In addition, representative force plots were constructed to illustrate the combined effects of gene-specific SHAP values in individual patients.

## 2.10 Immune cell infiltration analysis

Immune cell infiltration in IS and normal samples was quantified using MCPcounter [22], quanTIseq [23], xCell [24], and EPIC [25] algorithms. Differences between groups were evaluated with the Wilcoxon rank-sum test and visualized in a heatmap. Given that CLEC4E was identified as the most influential gene, its expression was further correlated with B- and T-cell infiltration across multiple algorithms using Spearman correlation analysis.

## 2.11 Functional analysis of CLEC4E

To explore the biological implications of CLEC4E in IS, we performed single-gene GSEA based on the KEGG pathway database. IS samples were stratified into high and low CLEC4E expression groups according to the median value. Enrichment analysis was conducted using the "clusterProfiler" package, with significance thresholds set at nominal $p < 0.05$ and $FDR < 0.25$. Pathways with high normalized enrichment scores (NES) were further examined to identify potential mechanisms linking CLEC4E to immune activation and inflammatory signaling in IS.

## 2.12 Cell culture and OGD/R model establishment

The human neuroblastoma cell line SH-SY5Y (CL-0208, Procell) was cultured in Dulbecco's Modified Eagle Medium (DMEM, Gibco, USA) supplemented with 10% fetal bovine serum (FBS, Gibco, USA) and 1% penicillin-streptomycin (Yeasen, China), and maintained in a humidified incubator at 37°C with 5% $CO_2$.

To establish an in vitro cell model mimicking cerebral ischemia-reperfusion injury, SH-SY5Y cells were rinsed twice with glucose-free DMEM, then transferred to glucose- and serum-free DMEM. The cells were incubated in a tri-gas hypoxic incubator under hypoxic conditions (95% $N_2$, 5% $CO_2$) at 37°C for 4 hours to induce oxygen-glucose deprivation (OGD). For reoxygenation (R), the culture medium was replaced with complete DMEM/F12 medium containing 10% FBS, and the cells were transferred back to a normoxic incubator (37°C, 5% $CO_2$) for subsequent culture. Cells in the control group were cultured in complete DMEM medium under normoxic conditions without OGD/R treatment throughout the experiment [26].

## 2.13 siRNA transfection

CLEC4E-targeting siRNA (si-CLEC4E) and negative control siRNA (si-NC) were both synthesized by GenePharma. Transfection was carried out using Lipofectamine™ 3000 according to the manufacturer's instructions. Cells at 60–70% confluence were transfected in Opti-MEM™ medium for 6 h, followed by medium replacement and additional 24 h incubation before OGD exposure. All siRNA sequences for CLEC4E are available in S1 Table.

## 2.14 Cell counting kit-8 (CCK-8) assay

SH-SY5Y cells were seeded in 96-well plates at a density of $1 \times 10^3$ cells per well; after cell treatment, 10 μL of CCK-8 reagent (Vazyme, China) was added to each well and incubated in a 37°C incubator for 2 h; a microplate reader was used to detect the absorbance at 450 nm, and the relative viability of cells in each group was calculated with reference to the cell viability of the Control group [27].

## 2.15 Cell apoptosis assay

Treated SH-SY5Y cells were collected and washed twice with pre-cooled PBS buffer. The cells were then resuspended in 100 μL of binding buffer, followed by the addition of 5 μL of Annexin V-FITC and 5 μL of propidium iodide (PI) (KeyGen

Biotech, China). The mixture was incubated at room temperature in the dark for 15 min, and the cell apoptosis rate was detected using a BD Biosciences flow cytometer.

## 2.16 ROS staining

After treatment, samples were incubated with 10 µM DCFH-DA (Beyotime, China) working solution (diluted in serum-free medium) at 37°C for 20 minutes. For tissue specimens, sufficient volume was used to ensure complete immersion (typically ≥ 1 mL per well in a six-well plate format). Following incubation, all samples were washed three times with serum-free medium to remove unbound probes. Cell nuclei were then counterstained with 5 µg/mL DAPI solution for 5 minutes. Fluorescence was visualized and imaged using a fluorescence microscope.

## 2.17 Macrophage polarization assay

Conditioned media from 4 groups of SH-SY5Y cells were collected and sterilized by filtration through a 0.22 µm membrane; THP-1-derived macrophages differentiated with 100 ng/mL PMA for 24 h were co-cultured with the filtered conditioned medium (conditioned medium: fresh complete medium = 1:1) for 24 h; qRT-PCR was used to detect the mRNA expression levels of macrophage polarization markers (CD86, IL-12, CD206, IL-10) and pro-inflammatory cytokines (TNF-α, IL-1β, IL-6).

## 2.18 RNA extraction and quantitative reverse transcription polymerase chain reaction (qRT-PCR)

Total RNA was extracted from cells and tissues using TRIzol reagent (Vazyme, China). Following extraction, 1 µg of RNA underwent reverse transcription to eliminate genomic DNA, according to the manufacturer's instructions provided with the reverse transcription kit (Takara, Japan). This process resulted in the synthesis of complementary DNA (cDNA). Subsequently, cDNA amplification was conducted using Fast Start Universal SYBR Green Master in a real-time PCR setup (Sigma-Aldrich, Germany). The relative quantification of the target gene was determined using the $2^{-\Delta\Delta Ct}$ method, with GAPDH employed as an internal control to normalize expression levels. This approach ensured accurate assessment of gene expression in the samples. Primers used are listed in S2 Table.

## 2.19 Constructing a mouse ischemia-reperfusion (I/R) model

All animal experimental procedures have been approved by the Institutional Animal Care and Use Committee of the Hubei Provincial Center for Disease Control and Prevention. Randomization was employed, with mice being assigned to the sham operation group, and the ischemia-reperfusion (I/R) group. The sample size was determined with reference to previous studies using similar MCAO/R models and routine practice in our laboratory, resulting in a final group size of 10 mice (n = 10). Firstly, the mice were anesthetized with pentobarbital sodium to expose the left common carotid artery, internal carotid artery, and external carotid artery. Then insert 7−0 silicone-coated monofilament nylon suture (RWD, MSMC21B120PK50, China) into the internal carotid artery and block the starting point of MCA. Remove the filament at 1 hour and establish an ischemia-reperfusion (I/R) model. The prespecified endpoint of this study was the assessment of histological and molecular biological indicators at 24 hours post-modeling. The mice were euthanized by tail vein injection of high-dose pentobarbital sodium 24 hours after modeling for tissue harvesting and subsequent experimental procedures. Throughout the process, the mice were in an anesthetized/unconscious state. Specifically, We first administered a sedative (5 mg/kg) via intraperitoneal injection. Once the animals entered a state of tranquility and were unconscious, we proceeded with an intravenous injection of an overdose of pentobarbital sodium (150 mg/kg). Subsequently, we monitored the heart rate, breathing, and pupillary reflex of the mice to confirm their death within a short period, which is used for tissue harvesting and subsequent experimental procedures. All subsequent histological evaluations (HE staining, Nissl staining) and image analyses were conducted under blinded conditions.

## 2.20 Hematoxylin and Eosin (HE) Staining

HE staining was utilized to assess the histological characteristics of brain tissues in the middle cerebral artery occlusion and reperfusion (MCAO/R) mouse model. Normal and injured brain tissues were fixed in 10% formalin, embedded in paraffin blocks, and sectioned into 5-micron-thick slices that were placed on microscope slides. After deparaffinization and rehydration, the sections were stained with hematoxylin to highlight cell nuclei. This was followed by counterstaining with eosin to enhance cytoplasmic visualization. The stained sections were examined under a microscope, and images were captured for subsequent analysis.

## 2.21 Nissl staining

Place the slides with the adhered sections in a constant temperature drying oven at 60°C for at least 30 minutes to 1 hour to ensure the tissues adhere tightly to the glass slides. Then, perform routine dewaxing to water (15 minutes each in xylene I and xylene II, followed by gradient dehydration with ethanol: 100% I, 100% II, 95%, 90%, 80%, 70%, and 50% for 5 minutes each). Rinse with distilled water three times, 5 minutes each time. Next, place the slides in a 60°C incubator and stain with 1% toluidine blue for 40 minutes (or with tar purple for 30 seconds). Quickly remove the slides and gently rinse away the excess dye with distilled water (for about 10 seconds). Subsequently, transfer the slides to distilled water for several washes (at least 3 times) to completely remove the acidic differentiating solution and stop the differentiation process, with each wash lasting about 10–20 seconds. After washing off the dye with distilled water, sequentially dehydrate in 70%, 80%, 95%, and 100% ethanol, and then clear with xylene. Finally, mount the slides with neutral resin.

## 2.22 Western blot

Total proteins were extracted from tissues utilizing RIPA lysis buffer, supplemented with 1% protease inhibitor cocktail tablets. The protein concentration was determined using a BCA assay kit, following a 30-minute incubation at 37°C. For analysis, 30 µg of protein samples were denatured at 95°C for 10 minutes and separated by SDS-PAGE on 10%–12% gels. Subsequently, the separated proteins were transferred to polyvinylidene difluoride (PVDF) membranes. Following a blocking step of 2 hours at room temperature using 5% nonfat milk, the membranes were incubated overnight at 4°C with specific primary antibodies of CLEC4E (1:500, PA5–116924, Invitrogen). Afterward, the membranes were incubated with appropriate secondary antibodies for 2 hours at room temperature. The protein bands were visualized using an enhanced chemiluminescence (ECL) detection system, and subsequent densitometric analysis was performed using e-BLOT software. GAPDH served as the internal control for normalization in the analysis.

## 2.23 Statistical analysis

Statistical analyses were conducted using R software (version 3.6.0). Differential gene screening and identification were performed using methods such as WGCNA, PPI network analysis, and Lasso Cox regression analysis. Additionally, ROC curve analysis and SHAP analysis were employed to determine key genes [28,29]. Intergroup comparisons were performed utilizing unpaired t-tests, Wilcoxon tests, or one-way analysis of variance (ANOVA) as appropriate. A p-value of less than 0.05 was considered statistically significant for all analyses [30].

## Results

### 3.1 ROS-related pathways are significantly upregulated in IS

We first evaluated the activity of five predefined ROS-related pathways in two independent cohorts (GSE58294 and GSE16561) using ssGSEA. Heatmap visualization revealed a consistent elevation of ROS pathway scores in IS samples compared with normal controls across both datasets (Fig 1A-1B). Subsequent differential analysis demonstrated that all five ROS pathways, including HALLMARK REACTIVE OXYGEN SPECIES PATHWAY, GOBP CELLULAR RESPONSE

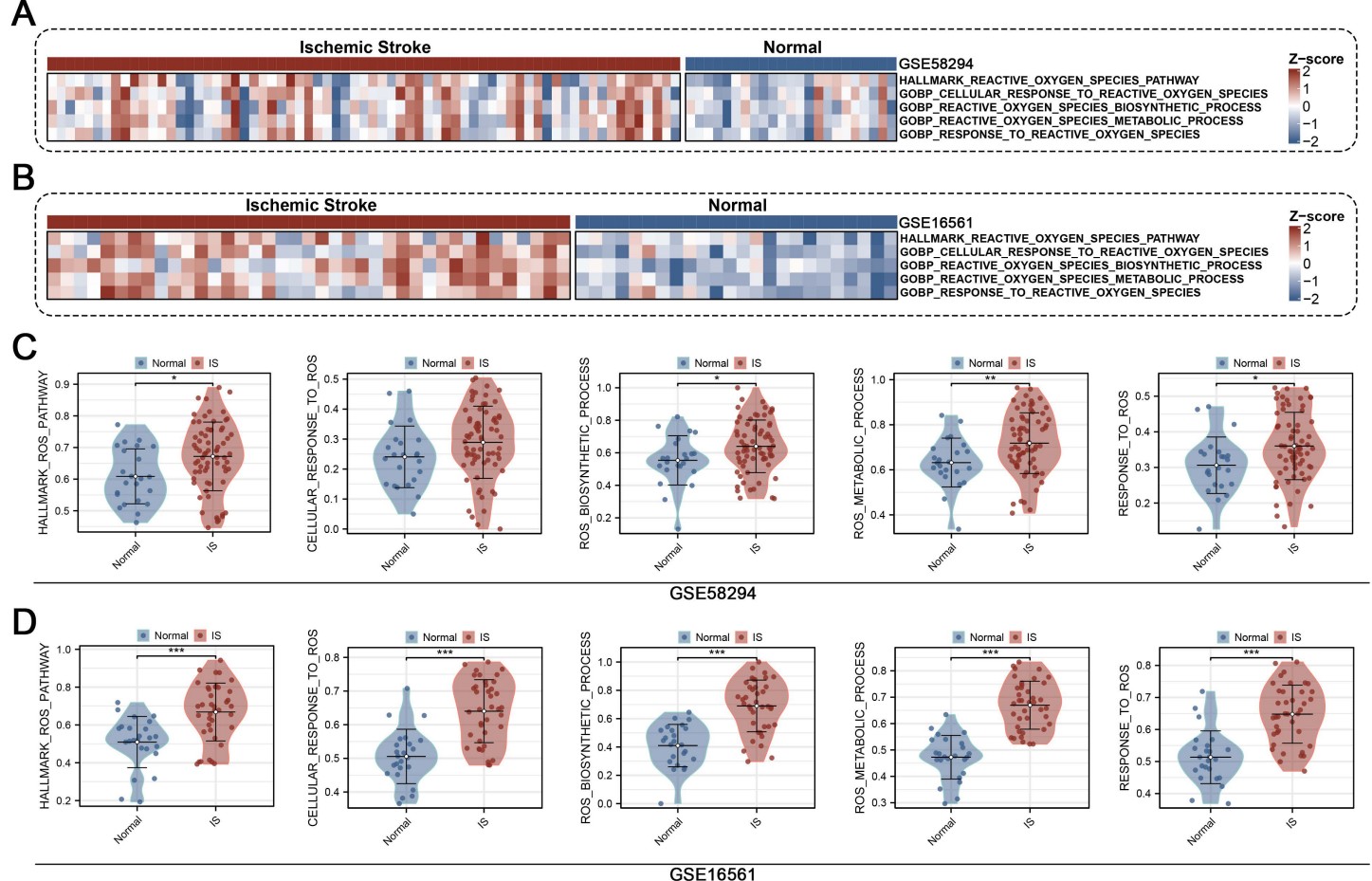

**Fig 1. ROS-related pathways are activated in IS.** (A) Heatmap of ssGSEA-derived scores for five ROS-related pathways in the GSE58294 cohort, showing higher enrichment in IS compared with normal controls. (B) Heatmap of pathway scores in the GSE16561 cohort, confirming consistent upregulation of ROS pathways in IS. (C) Violin plots of pathway activity in GSE58294, demonstrating significantly increased enrichment scores of ROS-related pathways in IS samples. (D) Violin plots of pathway activity in GSE16561, further validating the elevated ROS pathway activity in IS compared with controls.

TO REACTIVE OXYGEN SPECIES, GOBP REACTIVE OXYGEN SPECIES BIOSYNTHETIC PROCESS, GOBP REACTIVE OXYGEN SPECIES METABOLIC PROCESS, and GOBP RESPONSE TO REACTIVE OXYGEN SPECIES, were significantly enriched in IS patients (Fig 1C-1D). These results indicate a pervasive activation of ROS-associated biological programs in IS.

### 3.2 WGCNA reveals a ROS-associated co-expression module in IS

To delineate the co-expression patterns of ROS-related genes, we conducted WGCNA. The optimal soft-thresholding power was determined to ensure scale-free topology and stable network connectivity (Fig 2A). Hierarchical clustering grouped the genes into multiple modules with distinct co-expression patterns (Fig 2B). Subsequent module-trait correlation analysis demonstrated that several modules exhibited significant associations with ROS pathway activity, with the brown module showing the strongest and most consistent positive correlations across all five ROS-related pathways (Fig 2C). This finding suggests that the brown module captures a core gene set underlying ROS dysregulation in IS. To further

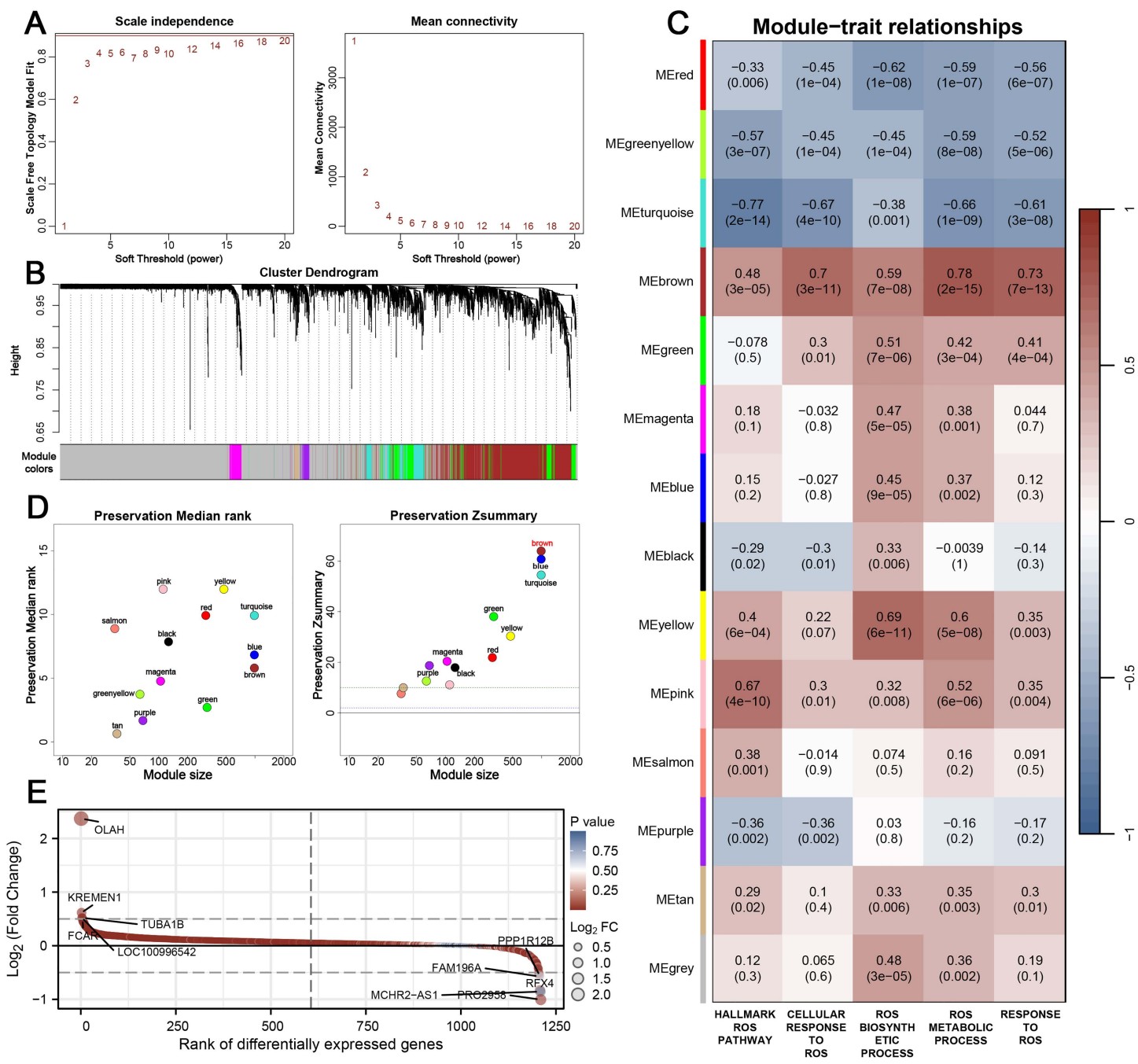

**Fig 2. Weighted gene co-expression network analysis identifies a ROS-related module in IS.** (A) Scale-free topology model fit and mean connectivity across a range of soft-thresholding powers. (B) Cluster dendrogram of ROS-related genes showing module assignment by color. (C) Heatmap of module-trait relationships depicting correlations between module eigengenes and five ROS-related pathways. The brown module showed the strongest positive associations with all pathways. (D) Module preservation analysis demonstrating high Z-summary and favorable median rank values for the brown module across datasets, supporting its stability and reproducibility. (E) Differential expression analysis of genes in the brown module, revealing significant dysregulation between IS and control samples.

validate the reliability of this module, preservation statistics were assessed across datasets. The brown module achieved high Z-summary scores and favorable median ranks, indicating strong reproducibility and stability of the co-expression pattern (Fig 2D). Differential expression analysis of genes within the brown module revealed that many were significantly dysregulated between IS and normal samples, underscoring their potential functional and clinical relevance (Fig 2E).

### 3.3 Biological functions and pathways of ROS-associated module genes

To gain further insight into the biological functions of ROS-associated genes, we analyzed the top 200 upregulated and 200 downregulated genes from the brown module. Construction of PPI networks revealed complex interactions among these genes, indicating their involvement in multiple coordinated biological processes (Fig 3A-3B). Gene Ontology enrichment analysis demonstrated that upregulated genes were predominantly enriched in immune-related processes such as myeloid leukocyte activation, leukocyte activation involved in immune response, and secretory granule organization, while also involving molecular functions including phospholipid binding and protein tyrosine kinase activity (Fig 3C). In contrast, down-regulated genes were associated with antigen processing and presentation, regulation of neuronal death, and transcriptional regulation, suggesting potential suppression of neuroprotective programs in IS (Fig 3D). KEGG pathway analysis further highlighted that upregulated genes were significantly enriched in platelet activation, Rap1 signaling, and Toll-like receptor signaling pathways (Fig 3E), while downregulated genes were mainly involved in bacterial invasion of epithelial cells, Fc gamma R-mediated phagocytosis, and endocytosis (Fig 3F). Collectively, these findings suggest that ROS-related gene dysregulation in IS is closely linked to immune activation, inflammatory signaling, and impaired cellular homeostasis.

### 3.4 Elastic net regression identifies key ROS-related genes predictive of IS

To establish a robust predictive model, we selected the top 10 upregulated and 10 downregulated genes from the brown module for Elastic Net regression analysis. Correlation analysis revealed significant co-expression relationships among these genes in the GSE58294 cohort, indicating potential functional interactions (Fig 4A). Using 10-fold cross-validation, the optimal λ value was determined, balancing model simplicity and prediction accuracy (Fig 4B-4C). The final model retained seven key genes, including CLEC4E, SLC8A1, HIST1H4H, BMX, MCEMP1, KREMEN1, and ZFP36L2, which demonstrated the strongest discriminatory power between IS and normal samples (Fig 4D). Model performance was further assessed in both the training dataset (GSE58294) and the independent validation cohort (GSE16561). In both datasets, the model achieved excellent separation between the IS and control groups (Fig 4E). These results highlight the predictive value of ROS-related gene signatures for IS.

### 3.5 Validation of predictive performance of ROS-related genes and risk score model

ROC analysis showed that the seven ROS-related genes exhibited strong discriminatory ability in the GSE58294 cohort, with AUC values ranging from 0.799 to 0.987, and maintained moderate performance in the GSE16561 cohort, with AUC values between 0.461 and 0.832 (Fig 5A-5B). To further evaluate the composite risk score model, which initially reached an AUC of 1.0 in GSE58294, suggesting possible overfitting, repeated cross-validation was conducted. The resulting out-of-fold AUC distributions demonstrated consistent robustness, with mean values of 0.81 in GSE58294 and 0.86 in GSE16561 (Fig 5C-5D). Calibration curves indicated excellent concordance between predicted and observed outcomes in GSE58294 (C-index = 0.995) and acceptable agreement in GSE16561 (C-index = 0.717) (Fig 5E-5F). Decision curve analysis further confirmed that the risk score model provided greater net clinical benefit than "treat all" or "treat none" strategies across a broad range of threshold probabilities in both datasets (Fig 5G-5H).

### 3.6 SHAP analysis interprets the contribution of ROS-related genes to the model

To enhance the interpretability of the predictive model, we applied SHAP analysis to quantify the contribution of each gene to the risk score. Feature importance ranking demonstrated that CLEC4E, MCEMP1, and HIST1H4H contributed the most

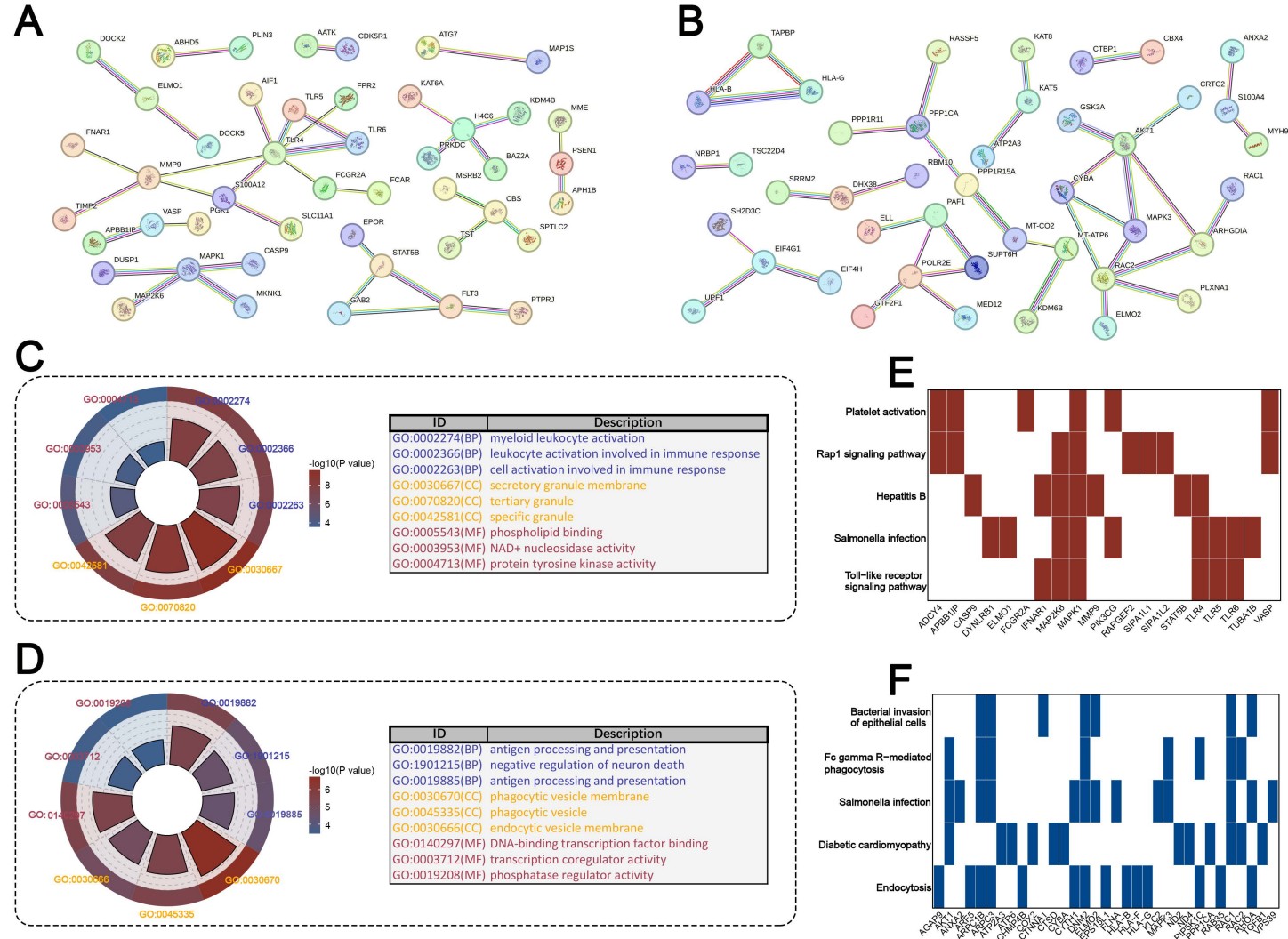

**Fig 3. Functional enrichment analysis of brown module genes in IS.** (A) Protein-protein interaction (PPI) network of the top 200 upregulated genes from the brown module, constructed using the STRING database. (B) PPI network of the top 200 downregulated brown module genes. (C) Gene Ontology (GO) enrichment analysis of upregulated genes, highlighting significant biological processes (BP), cellular components (CC), and molecular functions (MF). (D) GO enrichment analysis of downregulated genes, showing enrichment in antigen presentation, neuronal regulation, and transcriptional control. (E) KEGG pathway enrichment of upregulated genes, revealing strong associations with platelet activation, Rap1 signaling, Toll-like receptor signaling, and infection-related pathways. (F) KEGG enrichment of downregulated genes, indicating involvement in bacterial invasion of epithelial cells, Fc gamma R-mediated phagocytosis, endocytosis, and other cellular processes.

to model predictions, followed by SLC8A1, KREMEN1, BMX, and ZFP36L2 (Fig 6A). The summary plot further confirmed that higher expression of CLEC4E, MCEMP1, and HIST1H4H increased the predicted risk of IS, whereas ZFP36L2 exhibited an opposite effect (Fig 6B). Dependence plots showed consistent trends across individual genes, highlighting their nonlinear associations with model output and indicating both positive and negative contributions depending on expression level (Fig 6C). Finally, force plot visualization of a representative sample illustrated how gene-specific SHAP values combined to drive the final prediction, with elevated CLEC4E and MCEMP1 exerting strong positive contributions in IS, while ZFP36L2 provided a protective effect (Fig 6D). Collectively, these results underscore the biological plausibility and transparency of the seven-gene risk score model.

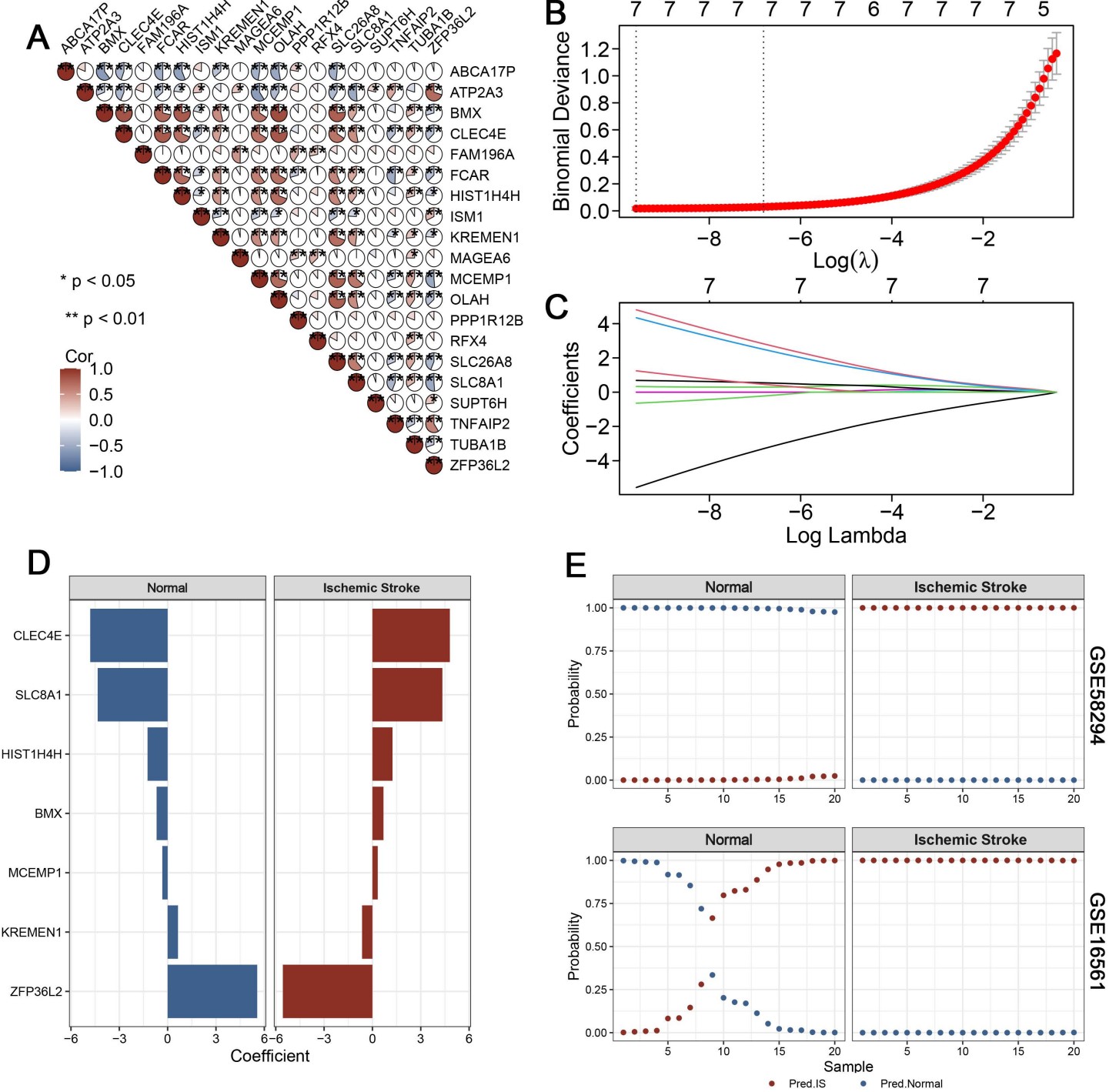

**Fig 4. Elastic Net regression identifies key ROS-related genes predictive of IS.** (A) Correlation matrix of the top 20 differentially expressed genes (10 upregulated and 10 downregulated) from the brown module in the GSE58294 cohort. (B) Ten-fold cross-validation results for binomial deviance across a range of λ values in Elastic Net regression. (C) Trajectories of regression coefficients for candidate genes as λ changes, showing progressive shrinkage of coefficients. (D) Bar plots displaying the coefficients of genes retained in the final Elastic Net model, highlighting seven key predictors of IS. (E) Model performance in the GSE58294 training set and independent GSE16561 validation cohort, showing clear separation of predicted probabilities between IS and normal samples.

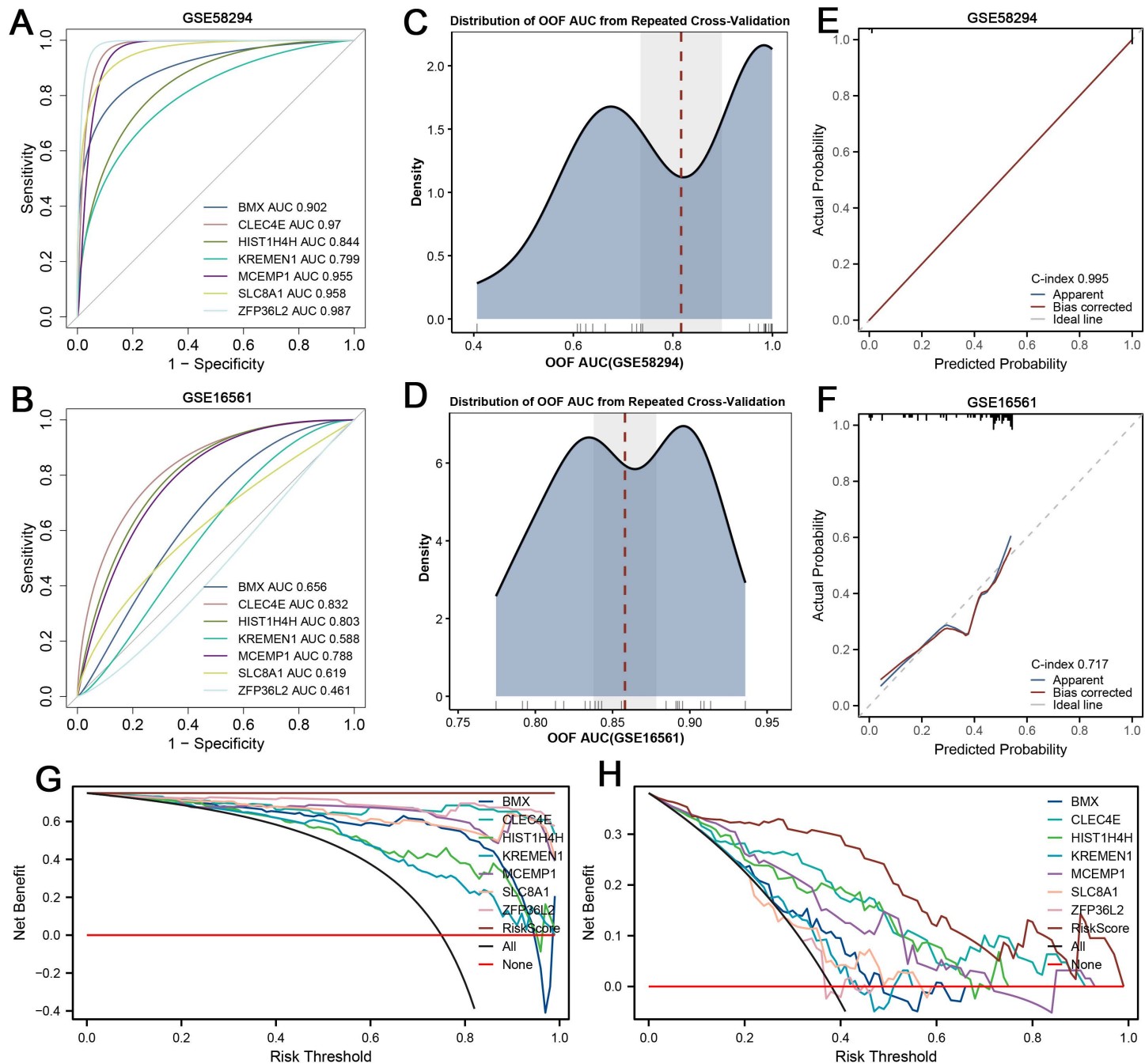

**Fig 5. Predictive performance of ROS-related genes and the risk score model.** (A-B) ROC curves of seven candidate genes in GSE58294 and GSE16561. (C-D) Out-of-fold (OOF) AUC distributions of the risk score from repeated cross-validation. (E-F) Calibration curves showing excellent agreement in GSE58294 (C-index=0.995) and acceptable agreement in GSE16561 (C-index=0.717). (G-H) Decision curve analysis (DCA) indicating a greater net clinical benefit of the risk score compared with "treat all" or "treat none" strategies in both cohorts.

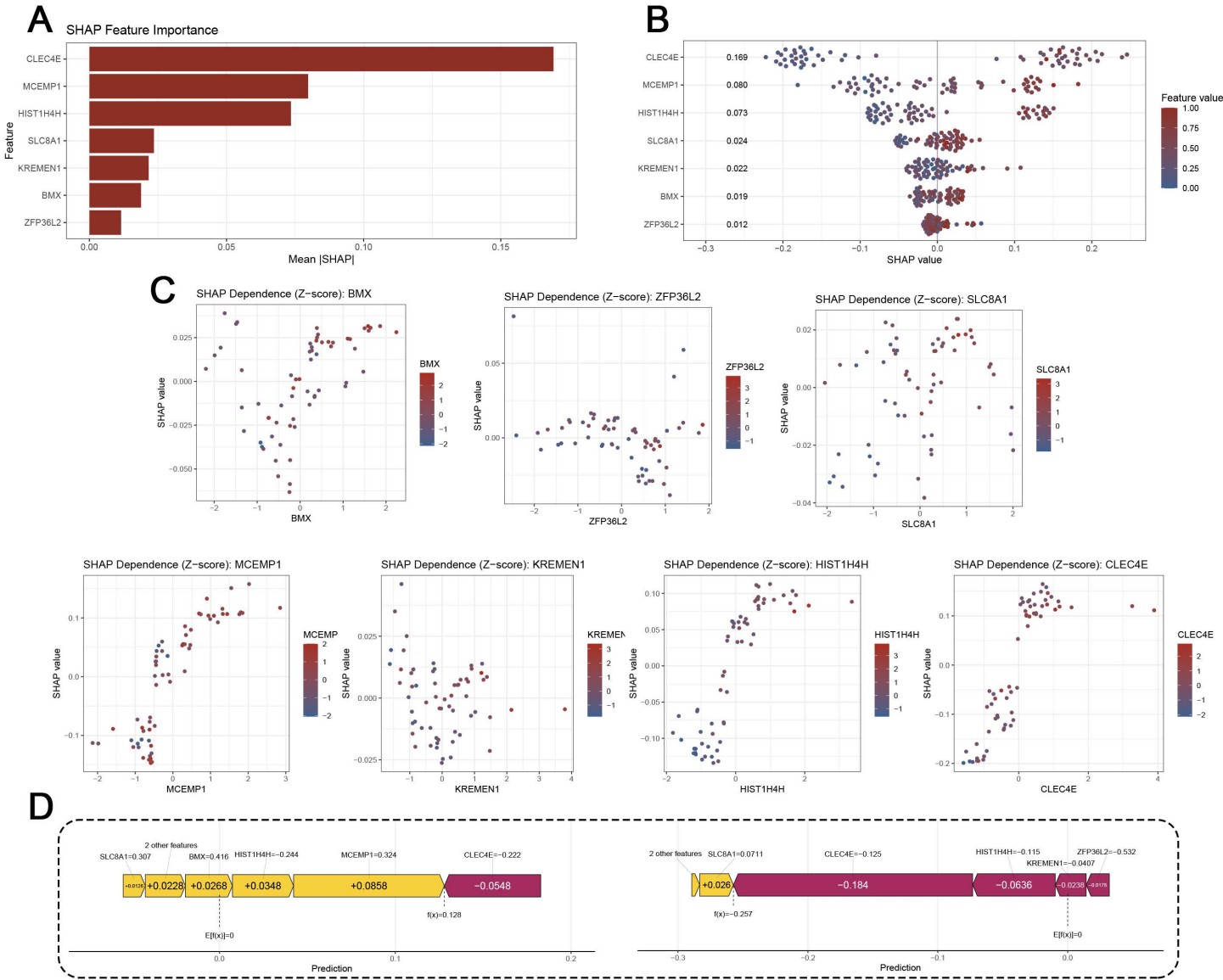

**Fig 6. SHAP analysis of gene contributions to the predictive model.** (A) SHAP feature importance ranking based on mean absolute SHAP values of the seven selected genes. (B) SHAP summary plot showing the distribution of SHAP values for each gene, with color indicating relative expression levels. (C) SHAP dependence plots displaying the relationship between gene expression (Z-score) and SHAP values for each of the seven genes. (D) Representative SHAP force plots illustrating the contribution of individual genes to the prediction in randomly selected normal and IS samples.

### 3.7 CLEC4E is associated with immune infiltration and inflammatory pathways in IS

Using multiple deconvolution algorithms, we systematically profiled immune infiltration patterns in IS and normal controls. Consistent results across MCPcounter, quanTIseq, xCell, and EPIC revealed significant alterations in several immune cell types, with B cells and T cells showing the most prominent differences between IS and normal groups (Fig 7A). Given its dominant contribution in the predictive model, we next assessed the relationship between CLEC4E and immune infiltration. Across algorithms, CLEC4E expression was consistently and negatively correlated with B-cell and T-cell abundance in IS samples (Fig 7B). This strong and consistent association positions CLEC4E as a potential indicator of an altered

immune microenvironment characterized by reduced lymphoid infiltration. To further investigate the functional implications of CLEC4E, we conducted a single-gene GSEA based on KEGG pathways in IS samples. The analysis demonstrated that high CLEC4E expression was significantly enriched in multiple inflammation- and immunity-related pathways, including MAPK signaling, RNA degradation, galactose metabolism, IL-5-mediated signaling, and neutrophil signaling (Fig 7C). These findings indicate that CLEC4E expression is associated not only with specific immune cell infiltration patterns but also with the enrichment of multiple inflammation-related signaling pathways in the context of ischemic stroke.

### 3.8 CLEC4E knockdown alleviates OGD/R-induced injury in SH-SY5Y cells and regulates macrophage polarization

To explore the role of CLEC4E in neuronal ischemia-reperfusion injury, we conducted experiments using the SH-SY5Y cell OGD/R model. qRT-PCR results showed that si-CLEC4E-1 had the highest knockdown efficiency among three si-CLEC4E sequences, so it was selected for subsequent experiments (Fig 8A). CCK-8 and flow cytometry assays demonstrated that OGD/R significantly reduced cell viability and increased apoptosis rate, while CLEC4E knockdown reversed these effects (Fig 8B-8C). ROS fluorescence staining revealed that CLEC4E knockdown inhibited OGD/R-induced ROS accumulation (Fig 8D). Additionally, co-culture experiments showed that conditioned medium from OGD/R-treated cells promoted macrophage M1 polarization (upregulating CD86, IL-12; downregulating CD206, IL-10) and increased pro-inflammatory cytokine expression (TNF-α, IL-1β, IL-6), which was reversed by CLEC4E knockdown (Fig 8E-8F). These results preliminarily indicate that CLEC4E knockdown alleviates OGD/R-induced neuronal injury by inhibiting oxidative stress and regulating macrophage polarization.

### 3.9 CLEC4E is significantly associated with ischemic stroke in vivo

To further investigate the role of the key gene CLEC4E in ischemic stroke, we established a middle cerebral artery occlusion and reperfusion (MCAO/R) model in mice. Histological analysis through HE staining revealed that the inflammatory damage in the IS group was significantly more severe compared to the control group (Fig 9A). Additionally, Nissl staining visually demonstrated that the extent of neuronal damage in the IS group was notably greater than that in the sham group (Fig 9B). At the molecular level, quantification through qRT-PCR and Western blot analysis revealed a significant increase in the expression of CLEC4E in the IS group, suggesting that CLEC4E may be closely associated with the inflammatory response and neuronal injury following cerebral ischemia. In addition, the expression of seven key genes in I/R models was confirmed to be consistent with our prediction (Fig 9C-9D). Furthermore, ROS immunofluorescence detection confirmed that the levels of ROS were elevated in the IS group (indicated by green fluorescence), indicating that the ischemia-reperfusion process triggered a more intense oxidative stress response (Fig 9E). These findings collectively imply that CLEC4E may contribute to the pathophysiological processes in cerebral ischemia by modulating inflammation and oxidative stress.

## 4. Discussion

In the present study, we systematically investigated the involvement of ROS-related pathways in IS by integrating transcriptomic datasets with multiple bioinformatics approaches. Our findings consistently demonstrated that ROS-related signaling was significantly upregulated in IS patients compared to healthy controls. Given that oxidative stress is one of the earliest and most sustained responses following cerebral ischemia, this observation reinforces the notion that dysregulated ROS metabolism is central to IS pathogenesis. The consistent upregulation across independent cohorts not only confirms robustness but also emphasizes that oxidative stress is a reproducible and indispensable hallmark of IS.

Through WGCNA analysis, we identified the Brown module as most strongly associated with ROS-related pathways. The tight correlation of this module with all five ROS signatures highlights its functional importance. Differential expression analysis further revealed that Brown module genes were significantly dysregulated in IS samples, suggesting that their

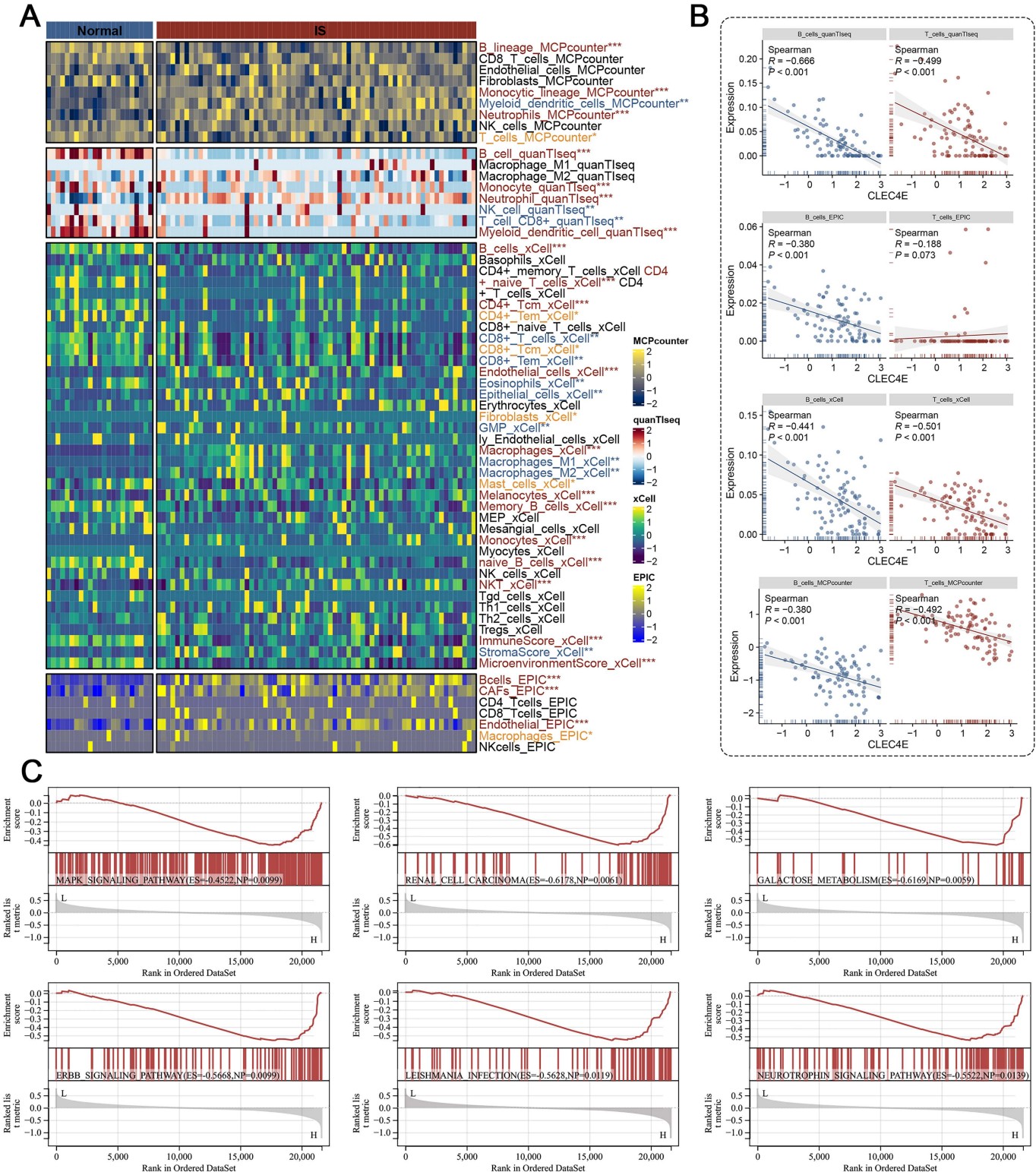

**Fig 7. Association of CLEC4E with immune infiltration and functional pathways in IS.** (A) Immune infiltration analysis using MCPcounter, quan-TIseq, xCell, and EPIC, showing differential immune cell abundances between IS and normal samples. (B) Correlation analysis of CLEC4E expression with B-cell and T-cell infiltration levels across multiple algorithms, indicating significant associations. (C) Single-gene GSEA of CLEC4E in IS samples,

demonstrating enrichment in inflammation- and immunity-related KEGG pathways, including MAPK signaling, RNA degradation, galactose metabolism, IL-5 mediated signaling, and neutrophil signaling.

altered expression may contribute to the observed enrichment of oxidative processes. Functional enrichment analysis provided additional insight, showing that these genes were predominantly involved in immune activation, cytokine signaling, and inflammatory responses. This supports the growing evidence that ROS does not merely cause cellular injury but also amplifies post-stroke neuroinflammation by regulating immune cell recruitment and activation [31,32].

Subsequent PPI network construction and enrichment analyses enabled the identification of hub genes that serve as potential key mediators linking ROS with immune dysregulation. Importantly, elastic net regression was employed to integrate these candidate genes into a diagnostic model. Unlike conventional regression methods, the elastic net is well-suited for handling high-dimensional transcriptomic data by balancing feature selection and collinearity control. The resulting gene signature exhibited outstanding predictive performance, with near-perfect AUCs in internal validation. To address potential overfitting, we further implemented OOF validation, which confirmed that the diagnostic model maintained high accuracy across resampled subsets. This strengthens confidence that the identified gene panel has translational potential as a biomarker set for IS.

Interpretability of machine learning models is often a critical barrier for clinical translation. By incorporating SHAP analysis, we quantified the contribution of each gene to the risk prediction. This approach not only validated the stability of the model but also provided biological plausibility by highlighting genes with established links to oxidative stress and immune regulation. Among these, CLEC4E was identified as a key gene in our predictive model and showed negative correlation with infiltration of B cells and T cells. Single-gene GSEA revealed that IS samples with high CLEC4E expression were significantly enriched in immune-related pathways such as MAPK signaling, IL-5-mediated signaling, and neutrophil signaling. These findings suggest that CLEC4E is not merely a biomarker but may actively participate in driving the vicious cycle of "oxidative stress–immune inflammation."

Experimental results confirmed that CLEC4E knockdown significantly alleviates oxygen-glucose deprivation/reoxygenation (OGD/R)-induced injury in SH-SY5Y neurons by inhibiting oxidative stress, regulating macrophage polarization, and reducing the release of pro-inflammatory cytokines; animal experiments further verified that in the mouse middle cerebral artery occlusion and reperfusion (MCAO/R) model, CLEC4E expression was significantly increased in the IS group, and was closely associated with inflammatory response, neuronal injury and enhanced oxidative stress after cerebral ischemia.

We propose a mechanistic model linking CLEC4E to ROS and immune infiltration: As a C-type lectin receptor, CLEC4E recognizes damage-associated molecules generated by ROS-mediated oxidation. Ligand binding activates MAPK and NF-κB pathways, promoting the expression of cytokines such as TNF-α, IL-1β, and IL-6. These cytokines further activate neutrophils and monocytes/macrophages, leading to a second wave of ROS production and forming a positive feedback loop. Concurrently, the negative correlation between CLEC4E expression and lymphocyte infiltration may result from CLEC4E-induced immunosuppressive microenvironment that suppresses lymphocyte function. Thus, CLEC4E acts as a molecular switch, converting early ROS signals into pro-inflammatory responses that exacerbate acute injury while potentially contributing to immunosuppressive networks that affect long-term recovery.

Traditional ischemic stroke biomarkers (e.g., NSE, S100B), which primarily reflect structural damage to neurons and glial cells, present several limitations: [1] insufficient sensitivity in the hyperacute phase, making early warning challenging; [2] inability to differentiate etiological subtypes of stroke; [3] limited to assessing injury severity without reflecting dynamic pathological processes such as immune inflammation. In contrast, the CLEC4E-based signature developed in this study dynamically captures the interactive state between oxidative stress and immune-inflammatory networks at the transcriptional level. This not only offers new potential for hyperacute diagnosis but also enables the identification of

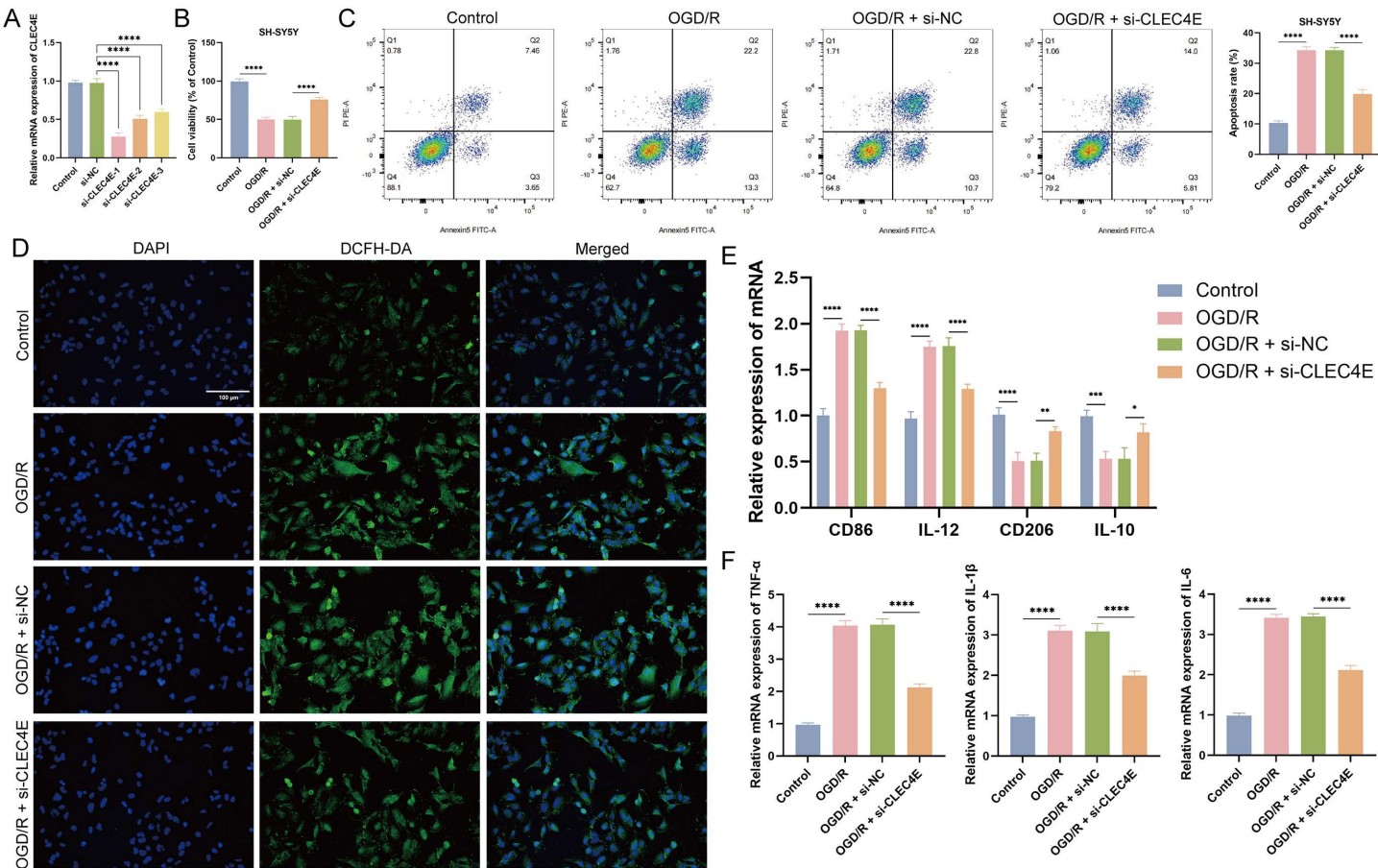

**Fig 8. Regulatory Effects of CLEC4E Knockdown on OGD/R Injury in SH-SY5Y Cells and Macrophage Polarization.** (A) Verification of knockdown efficiency of si-CLEC4E in SH-SY5Y cells (detected by qRT-PCR). (B) Effect of CLEC4E knockdown on OGD/R-induced changes in SH-SY5Y cell viability detected by CCK-8 assay. (C) Effect of CLEC4E knockdown on OGD/R-induced apoptosis of SH-SY5Y cells detected by flow cytometry. (D) Effect of CLEC4E knockdown on OGD/R-induced ROS levels in SH-SY5Y cells detected by immunofluorescence staining. (E) Effect of conditioned media from each group of SH-SY5Y cells on macrophage polarization (polarization markers detected by qRT-PCR). (F)Effect of conditioned media from each group of SH-SY5Y cells on the expression levels of pro-inflammatory cytokines in macrophages (detected by qRT-PCR). $*P < 0.05$, $**P < 0.01$, $***P < 0.001$, $****P < 0.0001$, t-test based $p$-value.

"high-oxidation, high-inflammation" patient subtypes through its scoring system, guiding personalized targeted therapies. More importantly, as a druggable cell-surface receptor, CLEC4E provides a direct target for developing monoclonal antibodies or small-molecule inhibitors, establishing a translational closed loop from diagnostic marker to therapeutic target.

These results extend previous reports implicating ROS in neuronal death, blood-brain barrier disruption, and sterile inflammation following ischemic injury [33,34]. By integrating ssGSEA, WGCNA, functional annotation, and machine learning, our study provides a gene-level perspective, identifying CLEC4E as a promising diagnostic marker and potential therapeutic target that bridges oxidative stress with immune dysregulation in IS.

Several limitations should be acknowledged. The study relied on retrospective transcriptomic datasets with limited clinical annotation, and although OOF validation reduced overfitting, prospective validation in independent cohorts remains necessary. The assumption of using blood-based signatures might not fully reflect the immune landscape in ischemic stroke tissue, which could be a potential area of future research. Moreover, the biological roles of CLEC4E and other hub genes were inferred computationally and require experimental confirmation. Finally, the use of transcriptomic data alone

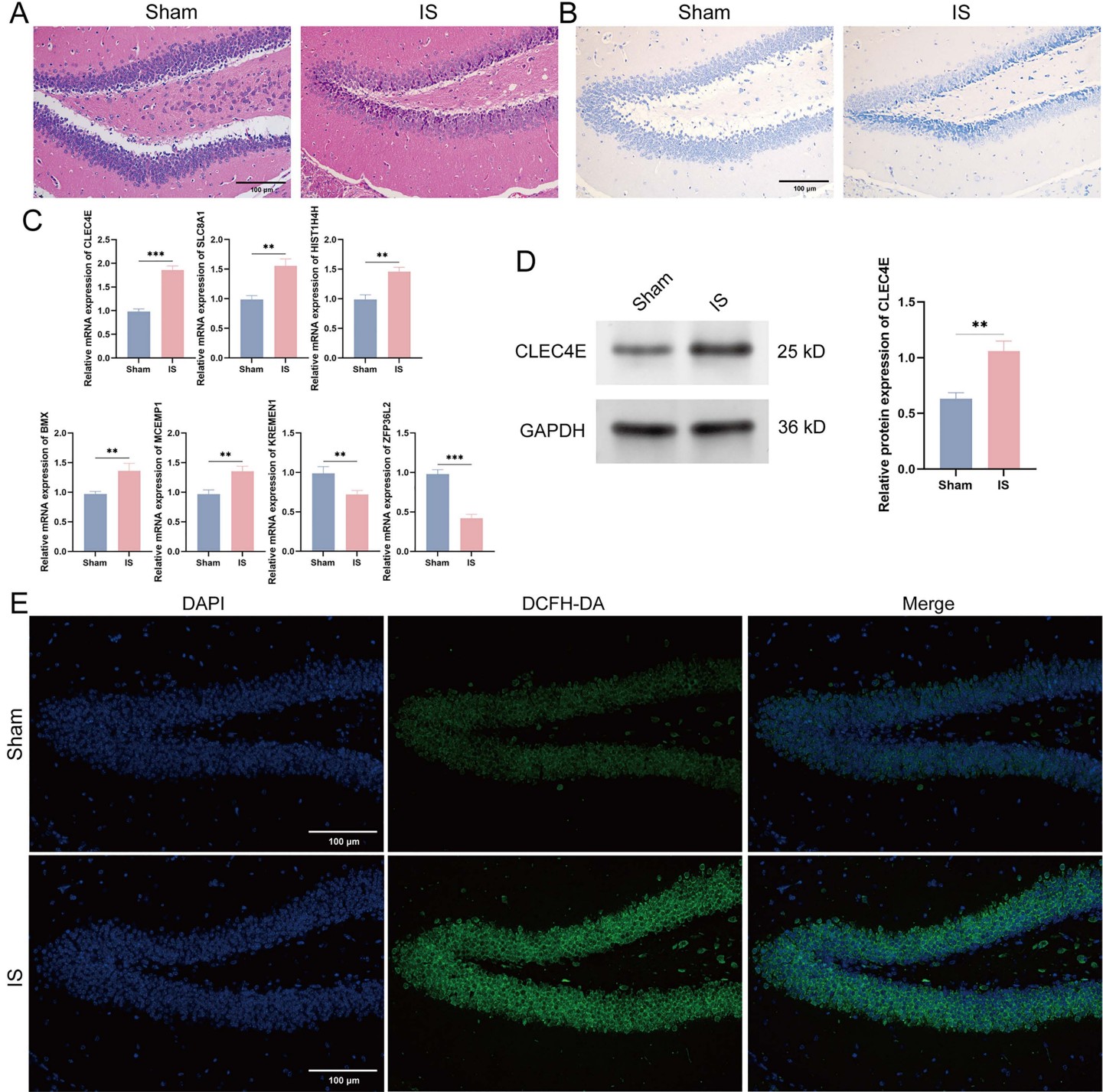

**Fig 9. Association of CLEC4E with IS in vivo.** (A) HE staining was used to assess the degree of brain inflammation in the sham and IS group. (B) Nissl staining was used to evaluate neuronal damage in the sham and IS group. (C) qRT-PCR was used to detect the mRNA expression levels of seven key genes in the sham and IS groups. (D) Western blot was used to detect the protein expression levels of CLEC4E in the sham and IS group. (E) ROS staining was used to detect the ROS levels in the sham and IS group. **$p<0.01$, ***$p<0.001$, t-test based $p$-value.

may not fully capture the complexity of IS, and future multi-omics integration will be needed to achieve a more comprehensive understanding.

In conclusion, our findings establish a strong link between ROS-related pathways and IS, identify a robust diagnostic gene signature, and highlight CLEC4E as a potential mediator of immune-oxidative interactions, thereby providing novel insights into IS pathogenesis and biomarker discovery.

## Supporting information

**S1 Table. siRNA sequences for CLEC4E.**
(DOCX)

**S2 Table. Primers for qRT-PCR. The code for all analysis.**
(DOCX)

**S3 File. Uncropped western blots.**
(PDF)

## Author contributions

**Conceptualization:** Xiaodi Ding.

**Data curation:** Lifang Yang, Tianyu Liang.

**Formal analysis:** Lifang Yang.

**Funding acquisition:** Lifang Yang.

**Investigation:** Lifang Yang, Tianyu Liang.

**Methodology:** Lifang Yang, Xiaodi Ding.

**Project administration:** Xiaodi Ding.

**Resources:** Xiaodi Ding.

**Software:** Lifang Yang.

**Supervision:** Xiaodi Ding.

**Validation:** Lifang Yang, Tianyu Liang.

**Visualization:** Lifang Yang, Tianyu Liang.

**Writing – original draft:** Lifang Yang, Xiaodi Ding.

**Writing – review & editing:** Lifang Yang, Tianyu Liang, Xiaodi Ding.

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
