## [Decision Letter · Decision Letter 0]

18 Dec 2025

Dear Dr. Ding,

Thank you for submitting your manuscript to PLOS ONE. After careful consideration, we feel that it has merit but does not fully meet PLOS ONE’s publication criteria as it currently stands. Therefore, we invite you to submit a revised version of the manuscript that addresses the points raised during the review process.

We look forward to receiving your revised manuscript.

Kind regards,

Yuzhen Xu

Academic Editor

PLOS One

2. To comply with PLOS One submissions requirements, in your Methods section, please provide additional information regarding the experiments involving animals and ensure you have included details on (1) methods of sacrifice, (2) methods of anesthesia and/or analgesia, and (3) efforts to alleviate suffering.

“This work was supported by the Zhejiang Provincial Administration of Traditional Chinese Medicine plans (Grant No. C-2025-W219).”

4. Thank you for stating the following in the Funding Section of your manuscript:

“This work was supported by the Zhejiang Provincial Administration of Traditional Chinese Medicine plans (Grant No. C-2025-W219).”

“This work was supported by the Zhejiang Provincial Administration of Traditional Chinese Medicine plans (Grant No. C-2025-W219).”

7. PLOS ONE now requires that authors provide the original uncropped and unadjusted images underlying all blot or gel results reported in a submission’s figures or Supporting Information files. This policy and the journal’s other requirements for blot/gel reporting and figure preparation are described in detail at https://journals.plos.org/plosone/s/figures#loc-blot-and-gel-reporting-requirements and https://journals.plos.org/plosone/s/figures#loc-preparing-figures-from-image-files. When you submit your revised manuscript, please ensure that your figures adhere fully to these guidelines and provide the original underlying images for all blot or gel data reported in your submission. See the following link for instructions on providing the original image data: https://journals.plos.org/plosone/s/figures#loc-original-images-for-blots-and-gels.

Reviewers' comments:

Reviewer's Responses to Questions

**Comments to the Author**

1. Is the manuscript technically sound, and do the data support the conclusions?

Reviewer #1: Yes

Reviewer #2: Yes

2. Has the statistical analysis been performed appropriately and rigorously?

Reviewer #1: Yes

Reviewer #2: Yes

3. Have the authors made all data underlying the findings in their manuscript fully available?

Reviewer #1: No

Reviewer #2: Yes

4. Is the manuscript presented in an intelligible fashion and written in standard English?

Reviewer #1: Yes

Reviewer #2: Yes

Reviewer #1: In this paper, we reveal the role of ROS-related gene features in ischemic stroke through systems biology methods, and propose CLEC4E as a potential immune regulator, which is innovative. However, there are some problems to be solved.

1. Supplementary method details

Specify the sample size of the cohort, quality control (PCA analysis), data preprocessing steps (such as batch effect correction, whether the data is preprocessed), and WGCNA module definition threshold (such as dynamic shear tree height).

2. The WGCNA in Figure 2 is an analysis using the GSE58294 or GSE16561 cohort. Please describe it clearly.

Which cohort was used for the LASSO regression analysis in Figure 4?

Please describe which cohort was used for the immune infiltration analysis in Figure 7?

3. The study fails to thoroughly explore how CLEC4E connects ROS with immune infiltration (e.g., receptor-ligand interactions and downstream signaling pathways). It is necessary to supplement the comparison with existing biomarkers (e.g., NSE and S100B).

4. The potential applications of gene signatures in early diagnosis, prognostic stratification, or therapeutic target development were not discussed. A comparison with existing biomarkers (e.g., NSE and S100B) should be supplemented.

5. Human Data Ethics Statement : The document fails to specify whether the GEO data has undergone ethical review. A supplementary statement regarding the ethical compliance of data sources should be included.

6. Completeness of supplementary materials : While uncropped Western blot images are labeled as "not for publication", critical experimental data should be made publicly available to ensure reproducibility. It is recommended that these key data be fully presented in the supplementary materials.

7. Please provide the code for all analysis.

Reviewer #2: This manuscript integrates two independent transcriptomic cohorts to map reactive oxygen species (ROS)–related programs in ischemic stroke and to derive a compact gene signature with diagnostic/predictive utility, complemented by immune-deconvolution analyses and in vivo validation.

1. Please provide a more explicit description of each GEO cohort (sample sizes, tissue/source, timing relative to stroke onset, key clinical covariates, and inclusion/exclusion criteria), as these factors can materially influence transcriptomic patterns and immune estimates. Also clarify how potential batch effects across platforms/cohorts were handled (or why not needed), how missing clinical information was operationalized, and whether any confounder adjustment was performed beyond quality control.

2. The analysis is anchored to five curated ROS-related gene sets; however, readers will benefit from a clear rationale for choosing these specific sets and how overlapping genes across sets were handled. Consider adding sensitivity analyses using alternative ROS resources (or broader oxidative-stress annotations) and report whether the key module and the final gene signature remain stable under reasonable variations in ROS gene set definitions.

3. Because WGCNA results can vary with parameter settings, please report key choices (soft-threshold selection criteria, minimum module size, merge thresholds) and provide more explicit module preservation outputs. It would also help to clarify whether the network was constructed on all genes vs. ROS-filtered genes (and the trade-offs), and to demonstrate that the “brown module” association is not driven by cohort-specific artifacts.

4. The manuscript would be strengthened by a clearer statement of the prediction target (diagnosis vs. risk vs. prognosis) and a consistent modeling description. Ensure that feature selection, model fitting, and hyperparameter tuning are fully nested within cross-validation to avoid optimistic bias. In addition to AUC, please report clinically relevant metrics (e.g., sensitivity/specificity at prespecified thresholds, PPV/NPV with plausible prevalence, and calibration diagnostics) and justify how decision-curve analysis was parameterized for this setting.

5. Deconvolution methods can diverge depending on signatures and tissue context. Please quantify concordance across algorithms, discuss assumptions (especially if derived from blood expression profiles), and consider additional checks (e.g., whether observed correlations persist after controlling for global inflammation markers). The negative association between CLEC4E and B/T cell infiltration is intriguing, but causal language should be avoided without mechanistic experiments.

6. It would be useful to briefly situate your pathway/functional enrichment pipeline within the landscape of contemporary large-cohort, reproducible enrichment workflows by adding a few representative citations (e.g., https://doi.org/10.1038/s41467-025-61785-z;
https://doi.org/10.1186/s12916-023-02920-9;
https://doi.org/10.1503/jpn.220202). This small addition would help readers benchmark your enrichment choices (gene-set resources, multiple-testing control, and interpretation strategy) against established population-scale practice.

7. In the Statistical analyses section, consider referencing several large-sample database studies that use closely related modeling and validation strategies (e.g., https://doi.org/10.1016/j.dcn.2025.101566;
https://doi.org/10.1016/j.psychres.2025.116503;
https://doi.org/10.1192/bjp.2023.107).

8. Please provide additional detail on randomization/blinding, sample sizes and power considerations, timepoints, and whether endpoints were prespecified. To better connect the computational findings to mechanism, consider functional perturbation (e.g., knockdown/overexpression or pharmacologic modulation) targeting CLEC4E (or its upstream/downstream signaling) and report whether this alters ROS burden, immune infiltration markers, and neuronal injury outcomes in the model.

**Do you want your identity to be public for this peer review?** For information about this choice, including consent withdrawal, please see our Privacy Policy

Reviewer #1: No

Reviewer #2: No

---

## [Author Response · Author response to Decision Letter 1]

9 Jan 2026

Manuscript Number: PONE-D-25-51535

Title: Systems Biology Analysis Uncovers a ROS-Associated Gene Signature and Immunomodulatory Role of CLEC4E in Ischemic Stroke

From:

Professor Xiaodi Ding

Center for Rehabilitation Medicine, Rehabilitation & Sports Medicine Research Institute of Zhejiang Province, Department of Rehabilitation Medicine, Zhejiang Provincial People’s Hospital(Affiliated People’s Hospital), Hangzhou Medical College, Hangzhou, 310014, Zhejiang, China.

January 9th, 2026

Dear Editor and reviewers:

Thank you very much for your detailed constructive and active comments on our manuscript.

According to each comment from reviewers, we have revised the manuscript carefully and made corresponding changes point by point. All the changes including the text and tables have been added and labeled in yellow in the revised manuscript according to this response. The list of changes has been itemized and explained in the response to reviewers.

Thanks again for your constructive suggestions. Thanks for your kind consideration of our article again. If you have any questions about the manuscript, please feel free to contact me. Merry Christmas!

Sincerely yours,

Professor Xiaodi Ding

E-mail: Qiwenlong11588@126.com

Responses to Journal requirements

Comment 1. Please ensure that your manuscript meets PLOS ONE's style requirements, including those for file naming. The PLOS ONE style templates can be found at https://journals.plos.org/plosone/s/file?id=wjVg/PLOSOne_formatting_sample_main_body.pdf and https://journals.plos.org/plosone/s/file?id=ba62/PLOSOne_formatting_sample_title_authors_affiliations.pdf

Response: Thank you for this valuable suggestion. We have made revisions in accordance with the style requirements of the PLOS ONE.

2. To comply with PLOS One submissions requirements, in your Methods section, please provide additional information regarding the experiments involving animals and ensure you have included details on (1) methods of sacrifice, (2) methods of anesthesia and/or analgesia, and (3) efforts to alleviate suffering.

Response: Thank you for this valuable suggestion. We have made detailed supplements to the animal experiment methods.

“This work was supported by the Zhejiang Provincial Administration of Traditional Chinese Medicine plans (Grant No. C-2025-W219).”

Response: Thank you for this valuable suggestion. “This work was supported by the Zhejiang Provincial Administration of Traditional Chinese Medicine plans (Grant No. C-2025-W219). The funders had no role in study design, data collection and analysis, decision to publish, or preparation of the manuscript.” We have included this amended Role of Funder statement in our cover letter.

4. Thank you for stating the following in the Funding Section of your manuscript:

“This work was supported by the Zhejiang Provincial Administration of Traditional Chinese Medicine plans (Grant No. C-2025-W219).”

“This work was supported by the Zhejiang Provincial Administration of Traditional Chinese Medicine plans (Grant No. C-2025-W219).”

Response: Thank you for this valuable suggestion. We have removed any funding-related text from the manuscript and included the amended Role of Funder statement in our cover letter.

“This work was supported by the Zhejiang Provincial Administration of Traditional Chinese Medicine plans (Grant No. C-2025-W219). The funders had no role in study design, data collection and analysis, decision to publish, or preparation of the manuscript.”

Response: Thank you for this valuable suggestion. I have linked my ORCID in the system.

Response: Thank you for this valuable suggestion. We have moved our ethics statement to the Methods section and delete it from any other section.

7. PLOS ONE now requires that authors provide the original uncropped and unadjusted images underlying all blot or gel results reported in a submission’s figures or Supporting Information files. This policy and the journal’s other requirements for blot/gel reporting and figure preparation are described in detail at https://journals.plos.org/plosone/s/figures#loc-blot-and-gel-reporting-requirements and https://journals.plos.org/plosone/s/figures#loc-preparing-figures-from-image-files. When you submit your revised manuscript, please ensure that your figures adhere fully to these guidelines and provide the original underlying images for all blot or gel data reported in your submission. See the following link for instructions on providing the original image data: https://journals.plos.org/plosone/s/figures#loc-original-images-for-blots-and-gels.

Response: Thank you for this valuable suggestion. We have uploaded the Uncropped western blots to the system as supplementary material as required.

Response: Thank you for this valuable suggestion. We have made the modifications as required.

Responses to Reviewer #1

In this paper, we reveal the role of ROS-related gene features in ischemic stroke through systems biology methods, and propose CLEC4E as a potential immune regulator, which is innovative. However, there are some problems to be solved.

Comment 1. Supplementary method details

Specify the sample size of the cohort, quality control (PCA analysis), data preprocessing steps (such as batch effect correction, whether the data is preprocessed), and WGCNA module definition threshold (such as dynamic shear tree height).

Response: Thank you for your valuable feedback. In response to your request, we have added the following key methodological details: (1). Sample sizes: The GSE58294 cohort includes 69 ischemic stroke (IS) samples and 23 normal controls; the GSE16561 cohort contains 39 IS samples and 24 normal controls. (2). Quality control and data preprocessing: Principal component analysis (PCA) was conducted for sample quality assessment. All samples exhibited acceptable clustering within their respective diagnostic groups, and no samples were excluded. Raw expression data were background-corrected and normalized using the limma package in R, with probes mapped to gene symbols based on platform annotation files. Since the two datasets were analyzed independently (e.g., within-cohort comparisons such as IS vs. control), systematic cross-dataset batch correction was not applied. 3.WGCNA parameter settings: A soft-thresholding power (β = 12) was selected to meet the scale-free topology criterion, which ensures the constructed network has biological rationality. Modules were identified through hierarchical clustering with the following parameters: a minimum module size of 30 genes, and a merge cut height of 0.25.

These details have been incorporated into Sections 2.1 and 2.4 of the manuscript, enhancing the transparency and reproducibility of our methodological approach.

Comment 2. The WGCNA in Figure 2 is an analysis using the GSE58294 or GSE16561 cohort. Please describe it clearly.

Which cohort was used for the LASSO regression analysis in Figure 4?

Please describe which cohort was used for the immune infiltration analysis in Figure 7?

Response: Thank you for raising these important points about cohort usage. We have provided the following clarifications: 1. The weighted gene co-expression network analysis (WGCNA) in Figure 2 was performed based on the GSE58294 cohort data. In this study, GSE58294 was used as the discovery set, and module preservation analysis was conducted via an internal sample stratification strategy (75% training set, 25% test set) to verify the robustness of co-expression modules. 2. The LASSO regression model (Figure 4) was trained on GSE58294 and independently validated on GSE16561. 3. Immune infiltration analysis (Figure 7) was performed on GSE58294 to maintain analytical consistency with the preceding steps.

These details have been integrated into the relevant method sections (2.4 and 2.7) to enhance clarity.

Comment 3. The study fails to thoroughly explore how CLEC4E connects ROS with immune infiltration (e.g., receptor-ligand interactions and downstream signaling pathways). It is necessary to supplement the comparison with existing biomarkers (e.g., NSE and S100B).

Response: Thank you for this valuable feedback. In response, we have expanded the Discussion section to address these points by: (1) detailing the potential molecular mechanisms linking CLEC4E to ROS‑immune crosstalk, including receptor‑ligand interactions and downstream signaling; and (2) providing a systematic comparison between our CLEC4E‑based signature and established biomarkers such as NSE and S100B.

Comment 4. The potential applications of gene signatures in early diagnosis, prognostic stratification, or therapeutic target development were not discussed. A comparison with existing biomarkers (e.g., NSE and S100B) should be supplemented.

Response: Thank you for this valuable suggestion. In response, we have expanded the Discussion to address clinical applications and comparisons with existing biomarkers:

Traditional ischemic stroke biomarkers (e.g., NSE, S100B), which primarily reflect structural damage to neurons and glial cells, present several limitations: (1) insufficient sensitivity in the hyperacute phase, making early warning challenging; (2) inability to differentiate etiological subtypes of stroke; (3) limited to assessing injury severity without reflecting dynamic pathological processes such as immune inflammation. In contrast, the CLEC4E-based signature developed in this study dynamically captures the interactive state between oxidative stress and immune-inflammatory networks at the transcriptional level. This not only offers new potential for hyperacute diagnosis but also enables the identification of “high-oxidation, high-inflammation” patient subtypes through its scoring system, guiding personalized targeted therapies. More importantly, as a druggable cell-surface receptor, CLEC4E provides a direct target for developing monoclonal antibodies or small-molecule inhibitors, establishing a translational closed loop from diagnostic marker to therapeutic target.

Comment 5. Human Data Ethics Statement: The document fails to specify whether the GEO data has undergone ethical review. A supplementary statement regarding the ethical compliance of data sources should be included.

Response: We thank the reviewer for raising this important point regarding ethical compliance. All data used in this study were obtained from the publicly available Gene Expression Omnibus (GEO) database (accession numbers: GSE58294 and GSE16561). As stated on the GEO platform and in the original publications citing these datasets, the data were collected in accordance with the ethical standards of the respective originating institutions, including obtaining informed consent from all participants and approval from the relevant institutional review boards (IRBs). A statement clarifying this has been added to the Methods section (Section 2.1).

Comment 6. Completeness of supplementary materials: While uncropped Western blot images are labeled as “not for publication”, critical experimental data should be made publicly available to ensure reproducibility. It is recommended that these key data be fully presented in the supplementary materials.

Response: We sincerely thank the reviewer for this important suggestion regarding data transparency. We have fully presented these key data in the supplementary materials. We believe these measures will significantly enhance the transparency and verifiability of our study.

Comment 7. Please provide the code for all analysis.

Response: Thank you for the suggestion. We have uploaded all the raw data to Science Data Bank (Data private access link: https://www.scidb.cn/en/s/zmeyMb), and the code for all analysis as a supplementary file upload system, to ensure full transparency and reproducibility.

Responses to Reviewer #2

This manuscript integrates two independent transcriptomic cohorts to map reactive oxygen species (ROS)–related programs in ischemic stroke and to derive a compact gene signature with diagnostic/predictive utility, complemented by immune-deconvolution analyses and in vivo validation.

Comment 1. Please provide a more explicit description of each GEO cohort (sample sizes, tissue/source, timing relative to stroke onset, key clinical covariates, and inclusion/exclusion criteria), as these factors can materially influence transcriptomic patterns and immune estimates. Also clarify how potential batch effects across platforms/cohorts were handled (or why not needed), how missing clinical information was operationalized, and whether any confounder adjustment was performed beyond quality control.

Response: Thank you for raising these important points regarding the detailed description of the GEO cohorts and our analytical approach. We appreciate the opportunity to clarify these methodological aspects. Below, we provide a summary of the relevant information based on the descriptions already included in the Methods section 2.1.

(1). GSE58294 contains peripheral blood samples from 69 ischemic stroke (IS) patients collected at three time points (within 3 hours, at 5 hours, and at 24 hours post-onset) and 23 healthy controls. GSE16561 consists of whole-blood samples from 39 IS patients and 24 healthy controls.

(2). Raw expression data from both cohorts were background-corrected and normalized using the limma package in R, with probes mapped to official gene s

---

## [Decision Letter · Decision Letter 1]

22 Feb 2026

Systems Biology Analysis Uncovers a ROS-Associated Gene Signature and Immunomodulatory Role of CLEC4E in Ischemic Stroke

PONE-D-25-51535R1

Dear Dr. Ding,

We’re pleased to inform you that your manuscript has been judged scientifically suitable for publication and will be formally accepted for publication once it meets all outstanding technical requirements.

Kind regards,

Yuzhen Xu

Academic Editor

PLOS One

Additional Editor Comments (optional):

I am pleased to inform you that your manuscript has been accepted following peer review.

Reviewers' comments:

Reviewer's Responses to Questions

**Comments to the Author**

Reviewer #1: All comments have been addressed

Reviewer #2: All comments have been addressed

Reviewer #3: All comments have been addressed

2. Is the manuscript technically sound, and do the data support the conclusions?

Reviewer #1: Yes

Reviewer #2: Yes

Reviewer #3: Yes

3. Has the statistical analysis been performed appropriately and rigorously?

Reviewer #1: Yes

Reviewer #2: Yes

Reviewer #3: Yes

4. Have the authors made all data underlying the findings in their manuscript fully available?

Reviewer #1: Yes

Reviewer #2: Yes

Reviewer #3: Yes

5. Is the manuscript presented in an intelligible fashion and written in standard English?

Reviewer #1: Yes

Reviewer #2: Yes

Reviewer #3: Yes

Reviewer #1: Great job! The author has thoroughly addressed all my questions and provided many solutions. I have no further questions.

Reviewer #2: The authors have satisfactorily addressed all of my previous comments and concerns. I am pleased with the revisions and recommend this manuscript for publication.

Reviewer #3: (No Response)

**Do you want your identity to be public for this peer review?** For information about this choice, including consent withdrawal, please see our Privacy Policy

Reviewer #1: No

Reviewer #2: No

Reviewer #3: No

---

## [Editor Report · Acceptance letter]

PONE-D-25-51535R1

PLOS One

Dear Dr. Ding,

I'm pleased to inform you that your manuscript has been deemed suitable for publication in PLOS One. Congratulations! Your manuscript is now being handed over to our production team.

Kind regards,

on behalf of

Professor Yuzhen Xu

Academic Editor

PLOS One